# CAT-Seg: Cost Aggregation for Open-vocabulary Semantic Segmentation

## Abstract

In this paper, we reinterpret the challenge of open-vocabulary semantic segmentation, where each pixel in an image is labeled with a wide range of text descriptions, as a correspondence problem focusing on the optimal text matching for each pixel. Addressing the limitations of conventional region-to-text matching approaches, we introduce a novel framework, CAT-Seg, grounded on the principles of cost aggregation methods in visual correspondence tasks. This framework refines the initial matching scores between dense image and text embeddings, leveraging a Transformer-based module for cost aggregation, further enhanced with embedding guidance. Notably, by operating on cosine similarity instead of manipulating embeddings directly, our approach enables the end-to-end fine-tuning of the CLIP model for pixel-level tasks, while yielding superior zero-shot capabilities. Empirical evaluations show our method's superior performance, achieving state-of-the-art results across open-vocabulary benchmarks, practical computational efficiency, and robustness for various domains, underscoring its potential for a wide range of open-vocabulary semantic segmentation applications.

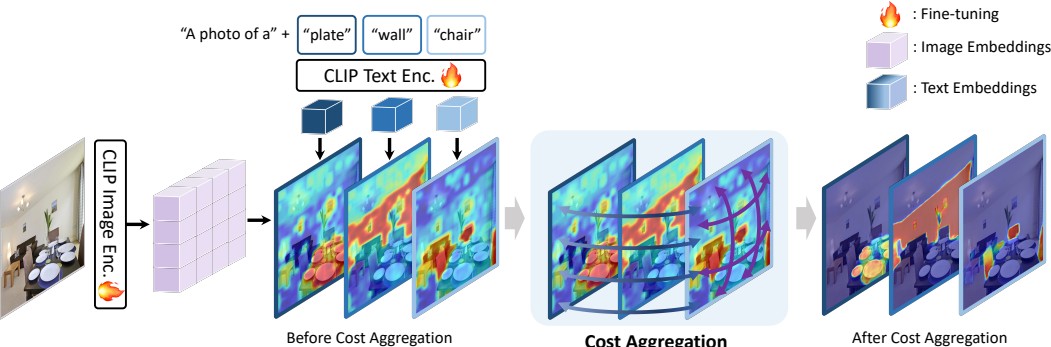

Figure 1: **We reformulate open-vocabulary semantic segmentation task to correspondence estimation task.** CAT-Seg constructs a cost volume from CLIP image and text embeddings and performs cost aggregation for its refinement, clearly setting a new state-of-the-art for all the standard benchmarks and an additional benchmark consisting of 22 datasets from various domains.

## 1 Introduction

Open-vocabulary semantic segmentation aims to label each pixel within an image with class labels, which assumes a wide-range of text description. This task presents significant challenges due to the nature of the dataset it requires, which involves pixel-level annotations with labels in a variety of natural language forms. Pixel-level annotation not only demands extensive annotation effort, but assigning a natural language description to each pixel is prohibitive. To address this, recent works have leveraged pre-trained vision-language foundational models, *e.g.*, CLIP (Radford et al., 2021) or ALIGN (Jia et al., 2021), trained with image-level contrastive learning. However, transferring the image-level representation to pixel-level task with a relatively small annotated dataset (Caesar et al., 2018) still remains as a primary concern for the many open-vocabulary semantic segmentation methods.

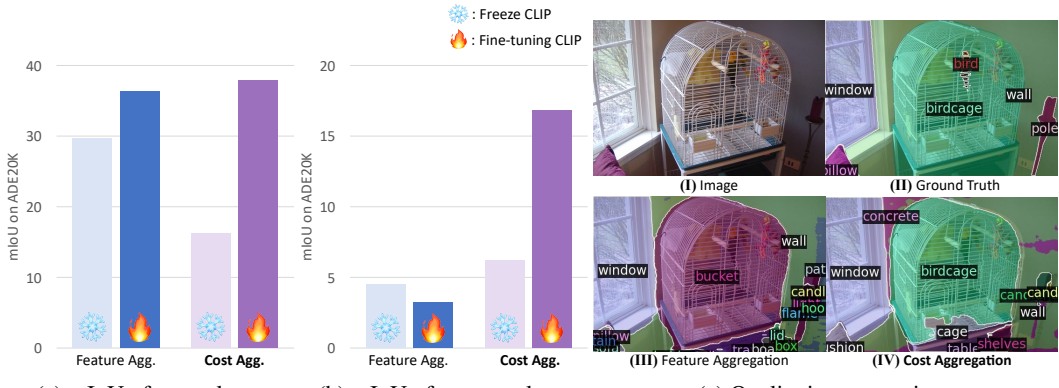

Figure 2: **Comparison between feature and cost aggregation.** To validate our framework, we examine outcomes of two approaches: feature aggregation, which directly processes dense image and text embeddings, and cost aggregation, which indirectly processes them through the use of a *cost map* encoding the dense similarity scores between dense image embeddings and text embeddings. From (a-b), unlike feature aggregation that suffers from a severe performance drop for unseen classes, cost aggregation achieves significant performance improvements upon fine-tuning of CLIP. This is also exemplified in qualitative results in (c). Our approach **(IV)** successfully segments the previously unseen class, such as "birdcage," whereas approach **(III)** fails.

Most prominent approach to solving this problem would be to transform it into a region-to-text matching problem. These approaches (Ding et al., 2022a; Ghiasi et al., 2022; Xu et al., 2022; Liang et al., 2022; Xu et al., 2023a;b; Yu et al., 2023) begin by grouping pixels into semantic regions with a class-agnostic mask generator, and then use CLIP to find a best corresponding class. However, the masks are generated independently from the classes provided by the user, limiting their adaptability to the virtually unlimited number of concepts that users can provide. Furthermore, since the mask generator is trained on a relatively small dataset compared to the image-caption dataset on which CLIP was trained, the mask generator tends to exhibit bias toward the training dataset (Liang et al., 2022). This motivates us to seek an alternative approach to generate masks conditioned on the provided classes, without relying to external mask generators.

In this work, we interpret the open-vocabulary semantic segmentation task as a correspondence problem that finds an optimal text matching for each pixel. In light of this perspective, we introduce a cost aggregation method (Kendall et al., 2017; Min & Cho, 2021), a well-established technique drawing from the visual correspondence task, which is known to be beneficial for generalization (Cai et al., 2020; Song et al., 2021; Liu et al., 2022). In visual correspondence, the cost aggregation methods first construct the initial matching score by computing pixel-to-pixel cosine similarity across the two images and sequentially refine the noisy initial score. Following our image-text correspondence formulation, we alter the method from addressing a pixel-to-pixel problem to addressing a pixel-to-text matching problem and designed a novel cost aggregator framework tailored to refine the image-text matching score.

The framework, dubbed CAT-Seg, is illustrated in Fig. 1. We initially compute a matching cost map between dense image and text embeddings using cosine similarity, which can also be interpreted as a rough segmentation map for a class. We then sequentially refine it within a cost aggregation stage, as the initial cost map contains noise and lacks fine details. Specifically, we employ a Transformer (Vaswani et al., 2017) based module that decomposes the aggregation process into spatial and class aggregation, aided by an additional technique called embedding guidance, to effectively aggregate the cost volume. Lastly, we use an upsampling decoder to upsample the aggregated cost volume while capturing fine details.

An important characteristic of our framework is its ability to handle CLIP's embedding space for leveraging its knowledge. Instead of directly manipulating the embeddings themselves, *e.g.* an additional mapping function (Zhou et al., 2022a), the framework conducts operations upon cosine similarity. Surprisingly, we empirically found that this design choice facilitates fine-tuning CLIP for pixel-level tasks, contrary to the previous observations that fine-tuning hindered its zero-shot

capability. A number of studies (Zhou et al., 2022a; Yu et al., 2023; Xu et al., 2023b) have reported that the fine-tuning approach fails because the encoders are prone to misalignment of the pre-trained image-text embedding space especially for unseen classes. As can be seen in Fig. 2, when the CLIP encoders are directly optimized on the feature representation, it fails to generalize to unseen classes. In contrast, when the encoders are fine-tuned within this framework, the CLIP model can be successfully adapted to the pixel-level while preserving its zero-shot capabilities.

By designing an aggregator tailored for processing the cost map and unveiling CLIP's dense zero-shot classification ability with its fine-tuning, the framework not only enables exceptional generalization ability but also demonstrates practical computational efficiency for both testing and training. We achieve state-of-the-art results on every standard open-vocabulary benchmark.

Furthermore, even in the extreme scenario (Blumenstiel et al., 2023) where the domain of the image and text description differs significantly from the training dataset, our model outperforms existing state-of-the-art methods with large margin, paving the way for domain-specific applications, such as paintings, body parts, engineering, or agriculture. This highlights versatility and potential of cost aggregation framework for practical open-vocabulary semantic segmentation.

## 2 RELATED WORK

**Open-vocabulary semantic segmentation.** Classical approaches to the task (Zhao et al., 2017; Bucher et al., 2019; Xian et al., 2019) attempt to learn visual embeddings that align with pre-defined text embeddings (Miller, 1998; Mikolov et al., 2013). However, the limited vocabulary of the words have been the major bottlenecks. To address this, LSeg (Li et al., 2022a) leveraged CLIP for learning pixel-level visual embeddings aligned with the text embeddings of CLIP. Alternatively, OpenSeg (Ghiasi et al., 2022) proposed to identify local regions within the image and correlate with the text embeddings with class-agnostic region proposals. Similarly, ZegFormer (Ding et al., 2022a) and ZSseg (Xu et al., 2022) proposed two-stage frameworks for dealing the task. Typically, they first learn to predict class-agnostic region proposals similar to (Ghiasi et al., 2022), and feed them to CLIP for final predictions. To better recognize these regions, OVSeg (Liang et al., 2022) collects region-text pairs to fine-tune the CLIP encoder, while MaskCLIP (Ding et al., 2022b) leverages the self-attention map from CLIP to refine the region proposals. Alternatively, ODISE (Xu et al., 2023a) leverages pre-trained Stable Diffusion (Rombach et al., 2022) model for generating high-quality class-agnostic masks. However, these region-to-text matching methods (Ding et al., 2022a; Ghiasi et al., 2022; Xu et al., 2022; Liang et al., 2022; Xu et al., 2023a;b; Yu et al., 2023) require a region generator, which is trained on a limited scale of annotated datasets.

More recently, ZegCLIP (Zhou et al., 2022e) and SAN (Xu et al., 2023b) proposed one-stage frameworks, where they attempts to leverage the embeddings from CLIP to predict masks instead of having class-agnostic mask generators parallel to CLIP. Although these methods can better leverage the pre-trained knowledge from CLIP, they introduce learnable tokens or adapter layers to the CLIP image encoder, which can be only trained on the seen classes. FC-CLIP (Yu et al., 2023) implements CLIP as the visual backbone for segmentation model, but opts for a frozen image encoder as they find fine-tuning the image encoder hinders performance for unseen classes. In contrast, we refrain from adding external layers within CLIP and achieve fine-tuning of the original encoders of CLIP by aggregating the cost volume, which is obtained solely from the embeddings of CLIP.

**Fine-tuning vision-language models.** Along with the advance of large-scale vision-language models, *e.g.* CLIP, numerous attempts have been made to adapt CLIP to various downstream tasks (Wortsman et al., 2022). CoOp (Zhou et al., 2022c) and CoCoOp (Zhou et al., 2022b) learns prompt tokens instead of optimizing the full model. Another stream of work is CLIP-Adapter Gao et al. (2023) and TIP-Adapter (Zhang et al., 2021), where they aggregate the image and text embeddings from CLIP through adapter layers instead of tuning the encoder itself. However, such methods mainly focus in few-shot settings rather than zero-shot evaluation. We explore end-to-end fine-tuning of CLIP for zero-shot pixel-level prediction, which has been failed in numerous attempts (Zhou et al., 2022a; Xu et al., 2023b; Yu et al., 2023).

**Cost aggregation.** Cost aggregation is a popular technique adopted for the process of establishing correspondence between visually or semantically similar images (Kendall et al., 2017; Guo et al., 2019; Yang et al., 2019; Cho et al., 2021; Hong et al., 2022a) by reducing the impact of errors and inconsistencies in the matching process. A matching cost, an input to cost aggregation, is typically

constructed between dense features extracted from a pair of images (Rocco et al., 2017), and often cosine-similarity (Liu et al., 2022; Rocco et al., 2017) is used. In matching literature, numerous works (Kendall et al., 2017; Chang & Chen, 2018; Guo et al., 2019; Yang et al., 2019; Song et al., 2021; Hong et al., 2022b; Huang et al., 2022; Cho et al., 2022) have proposed cost aggregation modules and demonstrated its importance, owing to its favorable generalization ability (Song et al., 2021; Liu et al., 2022). In this work, we leverage the cost volume constructed between image and text embeddings from CLIP encoders to promote accurate segmentation through cost aggregation.

## 3 METHODOLOGY

Given an image $I$ and a set of candidate class categories $\mathcal{C} = \{T(n)\}$ for $n = 1, \ldots, N_{\mathcal{C}}$, where $T(n)$ denotes textual description of $n$-th category and $N_{\mathcal{C}}$ is the number of classes, open-vocabulary semantic segmentation assigns a class label for each pixel in image $I$, where we interpret it as a correspondence problem between pixels in $I$ and a set of class labels $\mathcal{C}$. Different from classical semantic segmentation tasks (Long et al., 2015; He et al., 2017; Zhou et al., 2022d; He et al., 2019; Jin et al., 2021; Yu et al., 2020a; Yuan et al., 2020), open-vocabulary segmentation is additionally challenged by varying $\mathcal{C}$ at inference, which includes classes that were not observed during training.

Upon our formulation of solving open-vocabulary semantic segmentation as an pixel-to-text correspondence problem, we propose an framework for effectively finding such correspondence through cost aggregation (Kendall et al., 2017; Guo et al., 2019). We specifically design our approach for this task, which we call **C**ost **A**ggrega**T**ion approach for open-vocabulary semantic **Seg**mentation (CAT-Seg). This approach takes into consideration the handling of different modalities, *i.e.* image and text embeddings from CLIP, as we will describe in this section.

In Sec. 3.1, we provide an explanation of the cost computation and its embedding stage. In Sec. 3.2, we introduce a decomposed aggregation design tailored to address the pixel-to-text correspondence problem. This design is complemented by an embedding guidance that enhances the aggregation process. In Sec. 3.3, we demonstrate a decoder that upsamples the cost map while effectively handling fine details. In Sec. 3.4, we present an efficient fine-tuning technique for CLIP.

### 3.1 COST COMPUTATION AND EMBEDDING

Initially, we construct a cost map, which represents a matching score between dense image embeddings and text embeddings. To obtain dense CLIP embeddings, we follow the method described in (Zhou et al., 2022a), wherein we modify the last attention layer of the image encoder to eliminate the pooling effect. Given the modified CLIP image encoder $\Phi^V(\cdot)$ and the text encoder $\Phi^L(\cdot)$, we extract the dense image embeddings $D^V = \Phi^V(I) \in \mathbb{R}^{(H \times W) \times d}$ and the text embeddings $D^L = \Phi^L(T) \in \mathbb{R}^{N_c \times d}$, respectively. We use the image and text embeddings $D^V(i)$ and $D^L(n)$, where $i$ denotes 2D spatial positions of the image embedding and $n$ denotes an index for a class, to compute a cost volume $C \in \mathbb{R}^{(H \times W) \times N_c}$ by cosine similarity (Rocco et al., 2017). Formally, this is defined as:

$$C(i, n) = \frac{D^V(i) \cdot D^L(n)}{\|D^V(i)\|\|D^L(n)\|}. \tag{1}$$

To enhance the processing of cost in high dimensional feature space, we feed the cost volume to a single convolution layer that processes each cost slice $C(:, n) \in \mathbb{R}^{(H \times W) \times 1}$ independently to obtain initial cost volume embedding $F \in \mathbb{R}^{(H \times W) \times N_c \times d_F}$, where $d_F$ is the cost embedding dimension, as shown in Fig. 3.

### 3.2 COST AGGREGATION

To filter out noisy correlation within the cost volume, we feed the cost volume embedding to aggregation modules. A cost aggregation (Kendall et al., 2017), which was initially developed for the image correspondence problem and specifically designed to process an image-to-image correlation volume, does not need to account for modality differences. In contrast, as we address the challenge of image-to-text correspondence, we need to consider the multi-modality of the cost volume and the characteristics of each modality. This includes addressing aspects such as spatial smoothness or

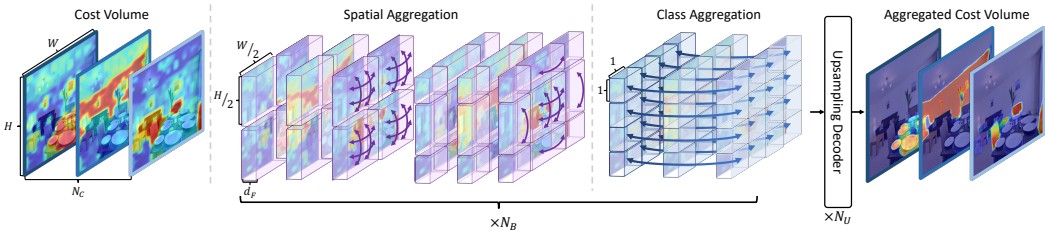

Figure 3: **Overview of the cost aggregation process.** Our cost aggregation consists of spatial aggregation and inter-class aggregation, followed by an upsampling decoder. Please refer to the supplementary material for a detailed illustration.

spatial relations in images, accounting for permutation invariance of classes, and accommodating the variable number of classes that may arise during inference.

In this regard, as shown in Fig. 3, we break down the aggregation stage into two separate module, *i.e.* spatial and class aggregation. Specifically, we perform spatial aggregation followed by class aggregation, and alternate both aggregations $N_B$ times. In addition, we facilitate the cost aggregation process with embedding guidance that provides contextual information from each modality. In the following, we explain each in detail.

**Spatial aggregation.**    Here, we reason about the spatial relations based on pixel-wise similarities computed between image and text embeddings. For this, we adopt Transformer (Vaswani et al., 2017; Liu et al., 2021) over CNNs for its adaptability to input tokens (Dai et al., 2021), while also having global (Vaswani et al., 2017) or semi-global (Liu et al., 2021; Hong et al., 2022a) receptive fields, which is more favorable to our goal to learn relations among all tokens. In practice, we employ Swin Transformer (Liu et al., 2021) for computational efficiency. We define this process as:

$$F'(:, n) = \mathcal{T}^{\text{sa}}(F(:, n)), \tag{2}$$

where $F(:, n) \in \mathbb{R}^{(H \times W) \times d_F}$, and $\mathcal{T}^{\text{sa}}(\cdot)$ denotes a pair of two consecutive Swin transformer block for spatial aggregation, where the first block features self-attention within a local window, followed by the second block with self-attention within shifted window. Note that we treat $d_F$ as channel dimensions for each token, and attention is computed within individual classes separately. Intuitively, we perform spatial aggregation for each class to locate the features that will guide to accurate segmentation outputs.

**Class aggregation.**    Subsequent to spatial aggregation, class aggregation is designed to explicitly capture relationships between different class categories. However, this task presents two challenges that need to be addressed: the variable number of categories $\mathcal{C}$ and their unordered input arrangement. To address these challenges, we employ a Transformer (Vaswani et al., 2017) model without position embedding for aggregation. This approach enables the handling of sequences of arbitrary length and provides the model with permutation invariance to inputs. This process is defined as:

$$F''(i, :) = \mathcal{T}^{\text{ca}}(F'(i, :)), \tag{3}$$

where $F'(i, :) \in \mathbb{R}^{N_C \times d_F}$, and $\mathcal{T}^{\text{ca}}(\cdot)$ denotes a transformer block for class aggregation. Although we can employ the same Swin Transformer (Liu et al., 2021) as for the spatial aggregation, we instead employ a linear transformer (Katharopoulos et al., 2020) as we do not need to consider spatial structure of the input tokens in this aggregation. Also, it offers a linear computational complexity with respect to the number of the tokens, allowing efficient computation.

**Embedding guidance.**    As a means to enhance cost aggregation process, we additionally leverage the embeddings $D_L$ and $D_V$ to provide spatial structure or contextual information of the inputs. Intuitively, we aim to guide the process with embeddings, based on the assumption that visually or semantically similar input tokens, e.g., color or category, have similar matching costs, inspired by cost volume filtering (Hosni et al., 2012; Sun et al., 2018) in stereo matching literature (Scharstein & Szeliski, 2002). Accordingly, we redefine Eq. 2 and Eq. 3 as:

$$\begin{aligned} F'(:, n) &= \mathcal{T}^{\text{sa}}([F(:, n); \mathcal{P}^V(D^V)]), \\ F''(i, :) &= \mathcal{T}^{\text{ca}}([F'(i, :); \mathcal{P}^L(D^L)]), \end{aligned} \tag{4}$$

where $[\cdot]$ denotes concatenation, $\mathcal{P}^V$ and $\mathcal{P}^L$ denote linear projection layer, $D^V \in \mathbb{R}^{(H \times W) \times d}$, and $D^L \in \mathbb{R}^{N_C \times d}$, where $d$ denotes the feature dimension. Notably, we only provide the embeddings to query and key as we find this is sufficient for embedding guidance.

Table 1: **Quantitative evaluation on in-domain datasets.** The best-performing results are presented in bold, while the second-best results are underlined. Improvements over the second-best methods are highlighted in green. †: Re-implemented version trained for full COCO-Stuff.

| Model | VLM | Additional Backbone | Training Dataset | Additional Dataset | A-847 | PC-459 | A-150 | PC-59 | PAS-20 | PAS-20$^b$ |
|---|---|---|---|---|---|---|---|---|---|---|
| SPNet (Xian et al., 2019) | - | ResNet-101 | PASCAL VOC | ✗ | - | - | - | 24.3 | 18.3 | - |
| ZS3Net (Bucher et al., 2019) | - | ResNet-101 | PASCAL VOC | ✗ | - | - | - | 19.4 | 38.3 | - |
| LSeg (Li et al., 2022a) | ViT-B/32 | ResNet-101 | PASCAL VOC-15 | ✗ | - | - | - | - | 47.4 | - |
| LSeg+ (Ghiasi et al., 2022) | ALIGN | ResNet-101 | COCO-Stuff | ✗ | 2.5 | 5.2 | 13.0 | 36.0 | - | 59.0 |
| ZegFormer (Ding et al., 2022a) | ViT-B/16 | ResNet-101 | COCO-Stuff-156 | ✗ | 4.9 | 9.1 | 16.9 | 42.8 | 86.2 | 62.7 |
| ZegFormer† (Ding et al., 2022a) | ViT-B/16 | ResNet-101 | COCO-Stuff | ✗ | 5.6 | 10.4 | 18.0 | 45.5 | 89.5 | 65.5 |
| ZSseg (Xu et al., 2022) | ViT-B/16 | ResNet-101 | COCO-Stuff | ✗ | 7.0 | - | 20.5 | 47.7 | 88.4 | - |
| OpenSeg (Ghiasi et al., 2022) | ALIGN | ResNet-101 | COCO Panoptic | ✓ | 4.4 | 7.9 | 17.5 | 40.1 | - | 63.8 |
| OVSeg (Liang et al., 2022) | ViT-B/16 | ResNet-101c | COCO-Stuff | ✓ | 7.1 | 11.0 | 24.8 | 53.3 | 92.6 | - |
| ZegCLIP (Zhou et al., 2022e) | ViT-B/16 | - | COCO-Stuff-156 | ✗ | - | - | - | 41.2 | 93.6 | - |
| SAN (Xu et al., 2023b) | ViT-B/16 | - | COCO-Stuff | ✗ | 10.1 | 12.6 | 27.5 | 53.8 | 94.0 | - |
| CAT-Seg (ours) | ViT-B/16 | - | COCO-Stuff | ✗ | **12.0** (+1.9) | **19.0** (+6.4) | **31.8** (+4.3) | **57.5** (+3.7) | **94.6** (+0.6) | **77.3** (+11.8) |
| LSeg (Li et al., 2022a) | ViT-B/32 | ViT-L/16 | PASCAL VOC-15 | ✗ | - | - | - | - | 52.3 | - |
| OpenSeg (Ghiasi et al., 2022) | ALIGN | Eff-B7 | COCO Panoptic | ✓ | 8.1 | 11.5 | 26.4 | 44.8 | - | 70.2 |
| OVSeg (Liang et al., 2022) | ViT-L/14 | Swin-B | COCO-Stuff | ✓ | 9.0 | 12.4 | 29.6 | 55.7 | 94.5 | - |
| SAN (Xu et al., 2023b) | ViT-L/14 | - | COCO-Stuff | ✗ | 12.4 | 15.7 | 32.1 | 57.7 | 94.6 | - |
| ODISE (Xu et al., 2023a) | ViT-L/14 | Stable Diffusion | COCO-Stuff | ✗ | 11.1 | 14.5 | 29.9 | 57.3 | - | - |
| CAT-Seg (ours) | ViT-L/14 | - | COCO-Stuff | ✗ | **16.0** (+3.6) | **23.8** (+8.1) | **37.9** (+5.8) | **63.3** (+5.6) | **97.0** (+2.4) | **82.5** (+12.3) |

## 3.3 UPSAMPLING DECODER

Given the aggregated cost volume, we aim to generate the final segmentation mask that captures fine-details via upsampling. The simplest approach would be using handcrafted upsamplers, *i.e.* bilinear upsampling, but we propose to conduct further aggregation within the decoder with light-weight convolution layers. Additionally, we provide a low-level feature map which acts as an effective guide to filter out the noises in the cost volume and exploit the higher-resolution spatial structure for preserving fine-details.

Specifically, we employ bilinear upsampling on the cost volume and concatenate it with the corresponding level of feature map, followed by a convolutional layer. We iterate this process $N_U$ times, generating a high-resolution output which is fed into the prediction head for final inference.

To extract the high-resolution feature map, we avoid using an additional feature backbone that would introduce heavy computation. Instead, similarly to Li et al. (2022b), we extract these maps from the CLIP image encoder. Specifically, we extract the feature map from the output of intermediate layers of CLIP ViT (Dosovitskiy et al., 2020) and then upsample them using a single layer transposed convolution. This approach allows us to efficiently leverage the well-learned representations of CLIP for upsampling. Please refer to the supplementary material for additional details.

## 3.4 EFFICIENT FINE-TUNING OF CLIP

Based on the observation in Fig. 2, we train our model in an end-to-end manner, including the image and text encoders of CLIP. However, fine-tuning the encoder, which can scale up to hundreds of millions of parameters, can be computationally expensive and memory-intensive. Additionally, freezing some of its layers may help CLIP preserve its original embedding space and boost its performance on pixel-level tasks (Zhou et al., 2022a). To this end, we extensively investigate which layers should be frozen within CLIP (Dosovitskiy et al., 2020) in Sec. 4.4.

Surprisingly, it turns out that fine-tuning only the query and value projections of CLIP ViT (Dosovitskiy et al., 2020) is sufficient to enable its transfer to dense tasks. Also, even though cost aggregation already effectively preserves the zero-shot capability of CLIP, this efficient fine-tuning further enhances our framework's efficiency and performance compared to full fine-tuning of CLIP.

## 4 EXPERIMENTS

## 4.1 DATASETS AND EVALUATION

We train our model on the COCO-Stuff (Caesar et al., 2018), which has 118k densely annotated training images with 171 categories, following (Liang et al., 2022). We employ the mean Intersection over Union (mIoU) as the evaluation metric for all experiments. For the evaluation, we conducted

Table 2: **Quantitative evaluation on MESS (Blumenstiel et al., 2023).** MESS includes a wide range of domain-specific datasets, which pose significant challenges due to their substantial domain differences from the training dataset. We report the average score for each domain. Please refer to the supplementary material for the results of all 22 datasets. *Random* is the result of uniform distributed prediction which represents the lower-bound, while *Best supervised* represents the upper-bound performance for the datasets.

| Model | VLM | Additional Backbone | General | Earth Monit. | Medical Sciences | Engineer. | Agri. and Biology | Mean |
|---|---|---|---|---|---|---|---|---|
| *Random (LB)* | - | - | *1.17* | *7.11* | *29.51* | *11.71* | *6.14* | *10.27* |
| *Best supervised (UB)* | - | - | *48.62* | *79.12* | *89.49* | *67.66* | *81.94* | *70.99* |
| ZSSeg (Xu et al., 2022) | ViT-B/16 | ResNet-101 | 19.98 | 17.98 | 41.82 | 14.0 | 22.32 | 22.73 |
| ZegFormer (Ding et al., 2022a) | ViT-B/16 | ResNet-101 | 13.57 | 17.25 | 17.47 | 17.92 | 25.78 | 17.57 |
| X-Decoder (Zou et al., 2023) | UniCL-T | Focal-T | 22.01 | 18.92 | 23.28 | 15.31 | 18.17 | 19.8 |
| OpenSeeD (Zhang et al., 2023) | UniCL-B | Swin-T | 22.49 | 25.11 | **44.44** | 16.5 | 10.35 | 24.33 |
| SAN (Xu et al., 2023b) | ViT-B/16 | - | 29.35 | 30.64 | 29.85 | **23.58** | 15.07 | 26.74 |
| CAT-Seg (ours) | ViT-B/16 | - | **38.69** (+9.34) | **35.91** (+5.27) | 28.09 (-16.35) | 20.34 (-3.24) | **32.57** (+6.79) | **31.96** (+5.22) |
| OVSeg (Liang et al., 2022) | ViT-L/14 | Swin-B | 29.54 | 29.04 | **31.9** | 14.16 | 28.64 | 26.94 |
| SAN (Xu et al., 2023b) | ViT-L/14 | - | 36.18 | 38.83 | 30.27 | 16.95 | 20.41 | 30.06 |
| CAT-Seg (ours) | ViT-L/14 | - | **44.69** (+8.51) | **39.99** (+1.16) | 24.70 (-7.2) | **20.20** (+3.25) | **38.61** (+9.97) | **34.70** (+4.64) |

experiments on two different sets of datasets (Zhou et al., 2019; Everingham et al., 2009; Mottaghi et al., 2014): a commonly used set of datasets following (Ghiasi et al., 2022), and a multi-domain evaluation set (Blumenstiel et al., 2023) containing domain-specific images and class labels.

**Datasets for in-domain evaluation.** For in-domain evalutaion, we evaluate our model on ADE20K (Zhou et al., 2019), PASCAL VOC (Everingham et al., 2009), and PASCAL-Context (Mottaghi et al., 2014) datasets. ADE20K has 20k training and 2k validation images, with two sets of categories: A-150 with 150 frequent classes and A-847 with 847 classes (Ding et al., 2022a). PASCAL-Context contains 5k training and validation images, with 459 classes in the full version (PC-459) and the most frequent 59 classes in the PC-59 version. PASCAL VOC has 20 object classes and a background class, with 1.5k training and validation images. We report PAS-20 using 20 object classes. We also report the score for PAS-20[b], which defines the "background" as classes present in PC-59 but not in PAS-20, as in Ghiasi et al. (2022).

**Datasets for multi-domain evaluation.** We conducted a multi-domain evaluation on the MESS benchmark (Blumenstiel et al., 2023), specifically designed to stress-test the real-world applicability of open-vocabulary models with 22 datasets. The benchmark includes a wide range of domain-specific datasets from fields such as earth monitoring, medical sciences, engineering, agriculture, and biology. Additionally, the benchmark contains a diverse set of general domains, encompassing driving scenes, maritime scenes, paintings, and body parts. We report the average scores for each domain in the main text for brevity. For the complete results and details of the 22 datasets, please refer to the supplementary material.

## 4.2 IMPLEMENTATION DETAILS

We train the CLIP image encoder and the cost aggregation module with per-pixel binary cross-entropy loss. We set $d_F = 128$, $N_B = 2$, $N_U = 2$ for all of our models. We implement our work using PyTorch (Paszke et al., 2019) and Detectron2 (Wu et al., 2019). AdamW (Loshchilov & Hutter, 2017) optimizer is used with a learning rate of $2 \cdot 10^{-4}$ for our model and $2 \cdot 10^{-6}$ for the CLIP, with weight decay set to $10^{-4}$. The batch size is set to 4. We use 4 NVIDIA RTX 3090 GPUs for training. All of the models are trained for 80k iterations, which takes 8 hours. Further details can be found in supplementary material. Our code will be made publicly available.

## 4.3 MAIN RESULTS

**Results of in-domain evaluation.** The evaluation of in-domain datasets are shown in Table 1. Overall, our method significantly outperforms all competing methods, including those (Ghiasi et al., 2022; Liang et al., 2022) that leverage additional datasets (Chen et al., 2015; Pont-Tuset et al., 2020) for further performance improvements. To ensure a fair comparison, we categorize the models based on the scale of the vision-language models (VLMs) they employ. First, we present results for models

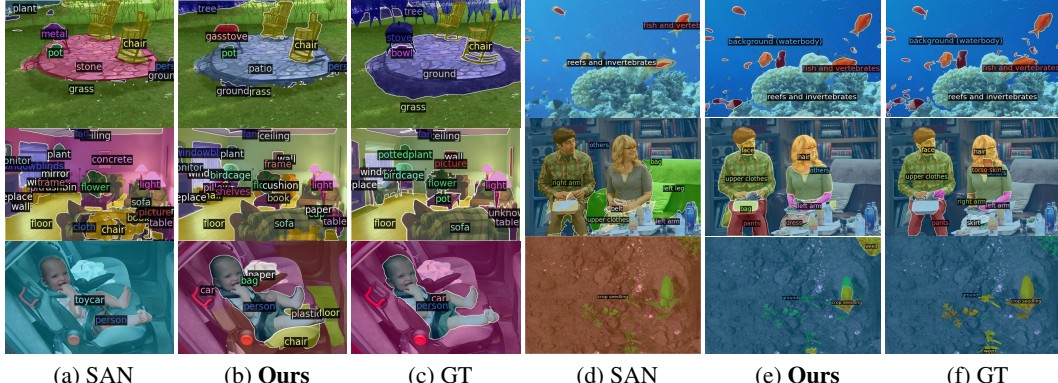

|        |        |        |        |        |        |
| ------ | ------ | ------ | ------ | ------ | ------ |
| (a) SAN | (b) **Ours** | (c) GT | (d) SAN | (e) **Ours** | (f) GT |

Figure 4: **Qualitative comparison to recent state-of-the-art (Xu et al., 2023b).** We visualize the results of PC-459 dataset in (a-c). For (d-f), we visualize the results from the MESS benchmark (Blumenstiel et al., 2023) across three domains: underwater (top), human parts (middle), and agriculture (bottom).

that use VLMs of comparable scale to ViT-B/16 (Dosovitskiy et al., 2020), and our model surpasses all previous methods, even achieving performance that matches or surpasses those using the ViT-L/14 model as their VLM (Xu et al., 2023b). For models employing the ViT-L/14 model as their VLM, our model demonstrates remarkable results, achieving a 16.0 mIoU in the challenging A-847 dataset and a 23.8 mIoU in PC-459. These results represent a 29% and 52% increase, respectively, compared to the previous state-of-the-art. We also present qualitative results of PASCAL-Context with 459 categories in Fig. 4, demonstrating the efficacy of our proposed approach in comparison to the current state-of-the-art methods (Ding et al., 2022a; Xu et al., 2022; Liang et al., 2022).

**Results of multi-domain evaluation.** In Table 2, we present the qualitative results obtained from the MESS benchmark (Blumenstiel et al., 2023). This benchmark assesses the real-world performance of a model across a wide range of domains. Notably, our model demonstrates a significant performance boost over other models, achieving the highest mean score. It particularly excels in the general domain as well as in agriculture and biology, showing its strong generalization ability. However, in the domains of medical sciences and engineering, the results exhibit inconsistencies with respect to the size of the VLM. Additionally, the scores for medical sciences are comparable to random predictions. We speculate that CLIP may have limited knowledge in these particular domains (Radford et al., 2021).

## 4.4 ANALYSIS AND ABLATION STUDY

**Component analysis.** Table 3 shows the effectiveness of the main components within our architecture through quantitative results. First, we introduce the baseline models in **(I)** and **(II)**, which simply feed the feature embeddings or the cost volume to the proposed upsampling decoder. We refer the readers to our supplementary materials for the details of the

Table 3: **Ablation study for CAT-Seg.** We conduct ablation study by gradually adding components to the cost aggregation baseline.

|        | Components | A-847 | PC-459 | A-150 | PC-59 | PAS-20 | PAS-20[b] |
| ------ | ---------- | ----- | ------ | ----- | ----- | ------ | ------ |
| **(I)** | Feature Agg. | 3.8 | 10.9 | 19.1 | 53.5 | 96.2 | 74.2 |
| **(II)** | Cost Agg. | 14.7 | 23.2 | 35.3 | 60.3 | 96.7 | 78.9 |
| **(III)** | **(II)** + Spatial agg. | 14.9 | 23.1 | 35.9 | 60.3 | 96.7 | 79.5 |
| **(IV)** | **(II)** + Class agg. | 14.7 | 21.5 | 36.6 | 60.6 | 95.5 | 80.5 |
| **(V)** | **(II)** + Spatial and Class agg. | 15.5 | 23.2 | 37.0 | 62.3 | 96.7 | 81.3 |
| **(VI)** | **(V)** + Embedding guidance | **16.0** | **23.8** | **37.9** | **63.3** | **97.0** | **82.5** |

baseline architecture. We first add the proposed spatial and class aggregations to the cost aggregation baseline in **(III)** and **(IV)**, respectively. In **(V)**, we interleave the spatial and class aggregations. Lastly, we add the proposed embedding guidance to **(V)**, which becomes our final model.

As shown, we stress the gap between **(I)** and **(II)**, which supports the findings presented in Fig. 2. Given that PAS-20 shares most of its classes with the training datasets(Xu et al., 2022), the performance gap between **(I)** and **(II)** is minor. However, for the challenging datasets such as A-847 or PC-459, the difference is notably significant, validating our cost aggregation framework for its generalizablity. We also highlight that as we incorporate the proposed spatial and class aggregation techniques, our approach **(V)** outperforms **(II)**, demonstrating the effectiveness of our design.

Finally, **(VI)** shows that our embedding guidance further improves performance across all the benchmarks.

**Analysis on fine-tuning of CLIP.** In this ablation study, we explore the various fine-tuning approach for the encoders of CLIP. In Table 4, we report the results of different approaches, which include the variant **(I)**: without fine-tuning, **(II)**: adopting Prompt Tuning (Zhou et al., 2022c; Jia et al., 2022), **(III)**: fine-tuning the entire CLIP, **(IV)**: fine-tuning the attention layer only (Touvron et al., 2022), **(V)**: fine-tuning query and key projections only, **(VI)**: fine-tuning key and value projections only, **(VII)**: our approach for CLIP image encoder only, **(VIII)**: our approach for text encoder only, and **(IX)**: our approach for the both encoders. Note that both image and text encoders are fine-tuned in **(I-VI)**. Overall, we observed that fine-tuning enhances the performance of our framework. Among the various fine-tuning methods, fine-tuning only the query and value projection yields the best performance improvement while also demonstrating high efficiency. Additionally, as can be seen in **(VII-IX)**, fine-tuning both encoders leads to better performance compared to fine-tuning only one of them in our framework.

Table 4: **Analysis of fine-tuning methods for CLIP.** We additionally note the number of learnable parameters of CLIP and memory consumption during training. Our method not only outperforms full fine-tuning, but also requires a smaller computational footprint.

|          | Methods        | A-847 | PC-459 | A-150 | PC-59 | PAS-20 | PAS-20$^b$ | #param. (M) | Memory (GiB) |
|----------|----------------|-------|--------|-------|-------|--------|-----------|-------------|--------------|
| **(I)**  | Freeze         | 4.8   | 8.7    | 20.8  | 52.5  | 84.5   | 76.7      | 5.8         | 20.0         |
| **(II)** | Prompt         | 8.8   | 14.3   | 30.5  | 55.8  | 93.2   | 74.7      | 7.0         | 20.9         |
| **(III)**| Full F.T.      | 13.6  | 22.2   | 34.0  | 61.1  | **97.3** | 79.7    | 393.2       | 26.8         |
| **(IV)** | Attn. F.T.     | 15.7  | 23.7   | 37.1  | 63.1  | 97.1   | 81.5      | 134.9       | 20.9         |
| **(V)**  | QK F.T.        | 15.3  | 23.0   | 36.3  | 62.0  | 95.9   | 81.9      | 70.3        | 20.9         |
| **(VI)** | KV F.T.        | **16.1** | **23.8** | 37.6 | 62.4 | 96.7 | 82.0 | 70.3 | 20.9 |
| **(VII)**| QV F.T. (Img.) | 13.9  | 22.8   | 35.1  | 62.0  | 96.3   | 82.0      | 56.7        | 20.9         |
| **(VIII)**| QV F.T. (Txt.)| 14.7  | 22.2   | 35.1  | 60.0  | 95.8   | 80.3      | 19.9        | 20.0         |
| **(IX)** | QV F.T. (Both) | 16.0  | **23.8** | **37.9** | **63.3** | 97.0 | **82.5** | 70.3 | 20.9 |

**Training with various datasets.** In this experiment, we further examine the generalization power of our method in comparison to other methods (Ding et al., 2022a; Xu et al., 2022) by training our model on smaller-scale datasets, which include A-150 and PC-59, that poses additional challenges to achieve good performance. The results are shown in Table 5. As shown, we find that although we observe some performance drops, which seem quite natural when a smaller dataset is used, our work significantly outperforms other competitors. These results highlight the strong generalization power of our framework, a favorable characteristic that suggests the practicality of our approach.

Table 5: **Training on various datasets.** CLIP with ViT-B is used for all methods. Our model demonstrates remarkable generalization capabilities even on relatively smaller datasets. The scores evaluated on the same dataset used for training are colored in gray.

| Methods        | Training dataset | A-847 | PC-459 | A-150 | PC-59 | PAS-20 | PAS-20$^b$ |
|----------------|------------------|-------|--------|-------|-------|--------|-----------|
| ZegFormer      | COCO-Stuff       | 5.6   | 10.4   | 18.0  | 45.5  | 89.5   | 65.5      |
| ZSseg          | COCO-Stuff       | 7.0   | 9.0    | 20.5  | 47.7  | 88.4   | 67.9      |
| CAT-Seg (ours) | COCO-Stuff       | **12.0** | **19.0** | **31.8** | **57.5** | **94.6** | **77.3** |
| ZegFormer      | A-150            | 6.8   | 7.1    | 33.1  | 34.7  | 77.2   | 53.6      |
| ZSseg          | A-150            | 7.6   | 7.1    | 40.3  | 39.7  | 80.9   | 61.1      |
| CAT-Seg (ours) | A-150            | **14.4** | **16.2** | 47.7 | **49.9** | **91.1** | **73.4** |
| ZegFormer      | PC-59            | 3.8   | 8.2    | 13.1  | 48.7  | 86.5   | 66.8      |
| ZSseg          | PC-59            | 3.0   | 7.6    | 11.9  | 54.7  | 87.7   | 71.7      |
| CAT-Seg (ours) | PC-59            | **9.6** | **16.7** | **27.4** | 63.7 | **93.5** | **79.9** |

**Efficiency comparison.** In Table 6, we thoroughly compare the efficiency of our method to recent methods (Ding et al., 2022a; Xu et al., 2022; Liang et al., 2022). We measure the number of learnable parameters, the total number of parameters, training time, inference time, and inference GFLOPs. Our model demonstrates strong efficiency in terms of both training and inference. This efficiency is achieved because our framework does not require an additional mask generator (Ding et al., 2022a).

Table 6: **Efficiency comparison.** All results are measured with a single RTX 3090 GPU.

| Methods                    | ZegFormer | ZSSeg   | OVSeg    | CAT-Seg (Ours) |
|----------------------------|-----------|---------|----------|----------------|
| # of learnable params. (M) | 103.3     | 102.8   | 408.9    | **70.3**       |
| # of total params. (M)     | 531.2     | 530.8   | 532.6    | **433.7**      |
| Training time (min)        | 1,148.3   | 958.5   | -        | **875.5**      |
| Inference time (s)         | 2.70      | 2.73    | 2.00     | **0.54**       |
| Inference GFLOPs           | 19,425.6  | 22,302.1 | 19,345.6 | **2,121.1**   |

## 5 CONCLUSION

In conclusion, our approach to open-vocabulary semantic segmentation reinterprets the task as an pixel-to-text correspondence challenge, effectively leveraging a tailored cost aggregation method. Through the introduction of our CAT-Seg framework, we harmonize the processing of CLIP's embedding space and its fine-tuning capabilities, enabling not only a superior generalization ability but also a practical computational efficiency. Our method surpasses previous state-of-the-arts in standard benchmarks, and also in scenarios with vast domain difference. The success in diverse domains, underscores the promise and potential of our cost aggregation framework in advancing the field of open-vocabulary semantic segmentation.

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

APPENDIX

In the following, we first provide more experimental results in Section A, followed by implementation details in Section B. We then provide additional analysis and ablation study in Section C. Finally, we present qualitative results for all the benchmarks and human part segmentation in Section D and a discussion of limitations in Section E.

## A   MORE RESULTS

Table 7: **Full results of quantitative evaluation on MESS (Blumenstiel et al., 2023).**

| | General | | | | | | Earth Monitoring | | | | | Medical Sciences | | | | Engineering | | | | Agri. and Biology | | | |
|---|---|---|---|---|---|---|---|---|---|---|---|---|---|---|---|---|---|---|---|---|---|---|---|
| | BDD100K | Dark Zurich | MHP v1 | FoodSeg103 | ATLANTIS | DRAM | iSAID | ISPRS Pots. | WorldFloods | FloodNet | UAVid | Kvasir-Inst. | CHASE DB1 | CryoNuSeg | PAXRay-4 | Corrosion CS | DeepCrack | PST900 | ZeroWaste-f | SUIM | CUB-200 | CWFID | Mean |
| *Random (LB)* | *1.48* | *1.31* | *1.27* | *0.23* | *0.56* | *2.16* | *0.56* | *8.02* | *18.43* | *3.39* | *5.18* | *27.99* | *27.25* | *31.25* | *31.53* | *9.3* | *26.52* | *4.52* | *6.49* | *5.3* | *0.06* | *13.08* | *10.27* |
| *Best sup. (UB)* | *44.8* | *63.9* | *50.0* | *45.1* | *42.22* | *45.71* | *65.3* | *87.56* | *92.71* | *82.22* | *67.8* | *93.7* | *97.05* | *73.45* | *93.77* | *49.92* | *85.9* | *82.3* | *52.5* | *74.0* | *84.6* | *87.23* | *70.99* |
| ZSSeg-B | 32.36 | 16.86 | 7.08 | 8.17 | 22.19 | 33.19 | 3.8 | 11.57 | 23.25 | 20.98 | 30.27 | 46.93 | 37.0 | **38.7** | 44.66 | 3.06 | 25.39 | 18.76 | 8.78 | 30.16 | 4.35 | 32.46 | 22.73 |
| ZegFormer-B | 14.14 | 4.52 | 4.33 | 10.01 | 18.98 | 29.45 | 2.68 | 14.04 | 25.93 | 22.74 | 20.84 | 27.39 | 12.47 | 11.94 | 18.09 | 4.78 | 29.77 | 19.63 | 17.52 | 28.28 | 16.8 | 32.26 | 17.57 |
| X-Decoder-T | 47.29 | 24.16 | 3.54 | 2.61 | 27.51 | 26.95 | 2.43 | 31.47 | 26.23 | 8.83 | 25.65 | 55.77 | 10.16 | 11.94 | 15.23 | 1.72 | 24.65 | 19.44 | 15.44 | 24.75 | 0.51 | 29.25 | 19.8 |
| SAN-B | 37.4 | 24.35 | 8.87 | 19.27 | 36.51 | 49.68 | 4.77 | 37.56 | 31.75 | 37.44 | 41.65 | 69.88 | 17.85 | 11.95 | 19.73 | 3.13 | 50.27 | 19.67 | 21.27 | 22.64 | 16.91 | 5.67 | 26.74 |
| OpenSeeD-T | 47.95 | 28.13 | 2.06 | 9.0 | 18.55 | 29.23 | 1.45 | 31.07 | 30.11 | 23.14 | 39.78 | 59.69 | 46.68 | 33.76 | 37.64 | 13.38 | 47.84 | 2.5 | 2.28 | 19.45 | 0.13 | 11.47 | 24.33 |
| Gr.-SAM-B | 41.58 | 20.91 | 29.38 | 10.48 | 17.33 | 57.38 | 12.22 | 26.68 | 33.41 | 19.19 | 38.34 | 46.82 | 23.56 | 38.06 | 41.07 | 20.88 | 59.02 | 21.39 | 16.74 | 14.13 | 0.43 | 38.41 | 28.52 |
| CAT-Seg-B | 46.71 | 28.86 | 23.74 | 26.69 | 40.31 | 65.81 | 19.34 | 45.36 | 35.72 | 37.57 | 41.55 | 48.2 | 16.99 | 15.7 | 31.48 | 12.29 | 31.67 | 19.88 | 17.52 | 44.71 | 10.23 | 42.77 | 31.96 |
| OVSeg-L | 45.28 | 22.53 | 6.24 | 16.43 | 33.44 | 53.33 | 8.28 | 31.03 | 31.48 | 35.59 | 38.8 | 71.13 | 20.95 | 13.45 | 22.06 | 6.82 | 16.22 | 21.89 | 11.71 | 38.17 | 14.0 | 33.76 | 26.94 |
| SAN-L | 43.81 | 30.39 | 9.34 | 24.46 | 40.66 | 68.44 | 11.77 | 51.45 | 48.24 | 39.26 | 43.41 | 72.18 | 7.64 | 11.94 | 29.33 | 6.83 | 23.65 | 19.01 | 18.32 | 40.01 | 19.3 | 1.91 | 30.06 |
| Gr.-SAM-L | 42.69 | 21.92 | 28.11 | 10.76 | 17.63 | 60.8 | 12.38 | 27.76 | 33.4 | 19.28 | 39.37 | 47.32 | 25.16 | 38.06 | 44.22 | 20.88 | 58.21 | 21.23 | 16.67 | 14.3 | 0.43 | 38.47 | 29.05 |
| CAT-Seg-L | 47.87 | 34.96 | 32.54 | 33.31 | 45.61 | 73.82 | 20.58 | 50.81 | 46.42 | 41.36 | 40.79 | 61.13 | 3.72 | 11.94 | 22.02 | 11.03 | 19.9 | 22.0 | 27.87 | 53.0 | 22.93 | 39.91 | 34.7 |

**Full quantitative results on MESS benchmark.**   In Table 7, we provide the results of all 22 datasets within MESS (Blumenstiel et al., 2023).

## B   MORE DETAILS

### B.1   ARCHITECTURAL DETAILS

In the following, we provide more architectural details. Our detailed overall architecture is illustrated in Fig. 5 (a).

**Embedding guidance.** In this paragraph, we provide more details of embedding guidance, which is designed to facilitate the cost aggregation process by exploiting its rich semantics for a guidance. We first extract visual and text embeddings from CLIP encoders (Radford et al., 2021). The embeddings then undergo linear projection and concatenated to the cost volume before query and key projections in aggregation layer. The design is illustrated in Fig. 5 (b).

**Upsampling decoder.** The detailed architecture is illustrated in Fig.5(c). In our upsampling decoder, we start by taking high-resolution features from the CLIP ViT model Dosovitskiy et al. (2020). We then apply a single transposed convolution layer to these extracted features to generate an upsampled feature map. Initially, the extracted feature maps have a resolution of $24 \times 24$ pixels. However, after processing them with the transposed convolution operation, we increase their resolution to $48 \times 48$ pixels for the first feature map, denoted as $E^V_{Dec,1}$, and to $96 \times 96$ pixels for the second feature map, denoted as $E^V_{Dec,2}$.

To obtain $E^V_{Dec,1}$, we utilize the output of the 8th layer for the ViT-B/16 model, and for the ViT-L/14 model, we use the output of the 16th layer. For the extraction of $E^V_{Dec,2}$, we employ shallower features: the output of the 4th layer for the ViT-B/16 model as a VLM, and the output of the 8th layer for the ViT-L/14 model. These features are employed to enhance cost embeddings with fine details using a U-Net-like architecture (Ronneberger et al., 2015).

### B.2   OTHER IMPLEMENTATION DETAILS

**Training details.** A resolution of $H = W = 24$ is used during training for constructing cost volume. The position embeddings of the CLIP image encoder is initialized with bicubic interpolation (Touvron et al., 2021), and we set training resolution as $384 \times 384$. For ViT-B and ViT-L variants,

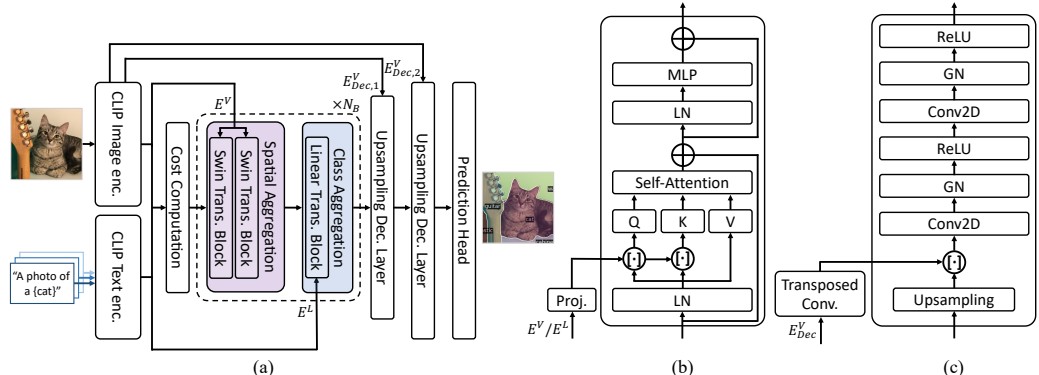

(a)  (b)  (c)

Figure 5: **More architectural details of CAT-Seg:** (a) overall architecture. (b) embedding guidance. Note that a generalized embedding guidance is illustrated to include different attention designs, *i.e.* shifted window attention (Liu et al., 2021) or linear attention (Katharopoulos et al., 2020). (c) upsampling decoder layer. GN: Group Normalization (Wu & He, 2018). LN: Layer Normalization (Ba et al., 2016).

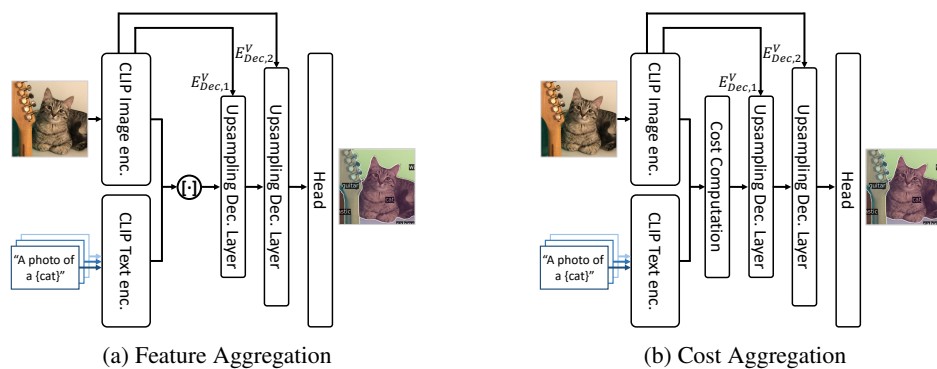

(a) Feature Aggregation  (b) Cost Aggregation

Figure 6: **Visualization of aggregation baselines.** (a) concatenates the features extracted from CLIP image and text encoders to feed into the upsampling decoder, while (b) constructs a cost volume using the image and text features from CLIP.

we initialize CLIP (Radford et al., 2021) with official weights of ViT-B/16 and ViT-L/14@336px respectively. All hyperparameters are kept constant across the evaluation datasets.

**Text prompt templates.** To obtain text embeddings from the text encoder, we form sentences with the class names, such as `"A photo of a {class}"`. We do not explore handcrafted prompts in this work, but it is open for future investigation.

**Feature and cost aggregation baselines.** In this paragraph, we provide more details of the architecture of two models introduced in Fig. 2: one is feature aggregation method and the other is cost aggregation method. As shown in Fig. 6 (a), the feature aggregation method directly leverages the features extracted from CLIP by feeding the concatenated image and text embeddings into the upsampling decoder. Fig. 6 (b) shows the cost aggregation approach that constructs cost volume instead, and subsequent embedding layer processes it to feed into upsampling decoder.

### B.3 PATCH INFERENCE

The practicality of Vision Transformer (ViT) (Dosovitskiy et al., 2020) for high-resolution image processing has been limited due to its quadratic complexity with respect to the sequence length. As our model leverages ViT to extract image embeddings, CAT-Seg may struggle to output to the conventional image resolutions commonly employed in semantic segmentation literature, such as $640 \times 640$ (Cheng et al., 2021; Ghiasi et al., 2022), without sacrificing some accuracy made by

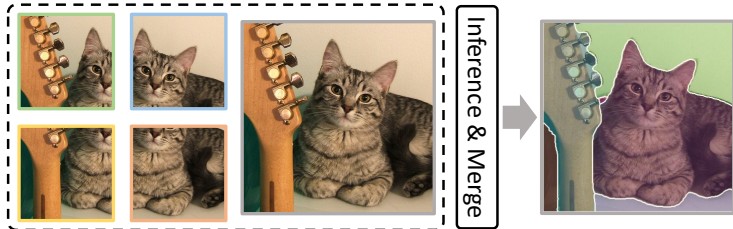

Figure 7: **Illustration of the patch inference.** During inference, we divide the input image into patches, thereby increasing the effective resolution.

losing some fine-details. Although we can adopt the same approach proposed in (Zhou et al., 2022a) to upsample the positional embedding (Zhou et al., 2022a), we ought to avoid introducing excessive computational burdens, and thus adopt an effective inference strategy without requiring additional training which is illustrated in Fig. 7.

To this end, we begin by partitioning the input image into overlapping patches of size $\frac{H}{N_P} \times \frac{W}{N_P}$. Intuitively, given an image size of $640 \times 640$, we partition the image to sub-images of size $384 \times 384$, which matches the image resolution at training phase, and each sub-images has overlapping regions $128 \times 128$. Subsequently, we feed these sub-images and the original image that is resized to $384 \times 384$ into the model. Given the results for each patches and the image, we merge the obtained prediction, while the overlapping regions are averaged to obtain the final prediction. In practice, we employ $N_P = 2$, while adjusting the overlapping region to match the effective resolution of $640 \times 640$.

## C  ADDITIONAL ABLATION STUDY

### C.1  ABLATION STUDY OF INFERENCE STRATEGY

| Methods | A-847 | PC-459 | A-150 | PC-59 | PAS-20 | PAS-20[b] |
|---|---|---|---|---|---|---|
| CAT-Seg w/ training reso. | 14.6 | 22.1 | 35.7 | 60.9 | 96.3 | 79.9 |
| CAT-Seg (ours) | **16.0** | **23.8** | **37.9** | **63.3** | **97.0** | **82.5** |

Table 8: **Ablation study of inference strategy.** CLIP with ViT-L is used for ablation.

Table 8 presents effects of different inference strategies for our model. The first row shows the results using the training resolution at inference time. The last row adopts the proposed patch inference strategy. It is shown that our proposed approach can bring large performance gains, compared to using the training resolution.

### C.2  EFFECTS OF UPSAMPLING DECODER

| Methods | A-847 | PC-459 | A-150 | PC-59 | PAS-20 | PAS-20[b] |
|---|---|---|---|---|---|---|
| CAT-Seg w/o upsampling decoder | 9.9 | 16.1 | 28.4 | 52.9 | 93.2 | 73.3 |
| CAT-Seg (ours) | **12.0** | **19.0** | **31.8** | **57.5** | **94.6** | **77.3** |

Table 9: **Ablation study of upsampling decoder.** CLIP with ViT-B is used for ablation.

We provide an quantitative results of adopting the proposed upsampling decoder in Table 9. The results show consistent improvements across all the benchmarks.

### C.3  MORE DETAILS OF MESS BENCHMARK

In Table 10, we provide details of the datasets in the MESS benchmark (Blumenstiel et al., 2023).

Table 10: **Full results of quantitative evaluation on mess (Blumenstiel et al., 2023).**

| Dataset | Link | Licence | Split | of classes | Classes |
|---|---|---|---|---|---|
| BDD100K (Yu et al., 2020b) | berkeley.edu | custom | val | 19 | [road; sidewalk; building; wall; fence; pole; traffic light; traffic sign; ...] |
| Dark Zurich (Sakaridis et al., 2019) | ethz.ch | custom | val | 20 | [unlabeled; road; sidewalk; building; wall; fence; pole; traffic light; ...] |
| MHP v1 (Li et al., 2017) | github.com | custom | test | 19 | [others; hat; hair; sunglasses; upper clothes; skirt; pants; dress; ...] |
| FoodSeg103 (Wu et al., 2021) | github.io | Apache 2.0 | test | 104 | [background; candy; egg tart; french fries; chocolate; biscuit; popcorn; ...] |
| ATLANTIS (Erfani et al., 2022) | github.com | Flickr (images) | test | 56 | [bicycle; boat; breakwater; bridge; building; bus; canal; car; ...] |
| DRAM (Cohen et al., 2022) | ac.il | custom (in download) | test | 12 | [bird; boat; bottle; cat; chair; cow; dog; horse; ...] |
| iSAID (Waqas Zamir et al., 2019) | github.io | Google Earth (images) | val | 16 | [others; boat; storage tank; baseball diamond; tennis court; bridge; ...] |
| ISPRS Potsdam (BSF Swissphoto, 2012) | isprs.org | no licence provided[1] | test | 6 | [road; building; grass; tree; car; others] |
| WorldFloods (Mateo-Garcia et al., 2021) | github.com | CC NC 4.0 | test | 3 | [land; water and flood; cloud] |
| FloodNet (Rahnemoonfar et al., 2021) | github.com | custom | test | 10 | [building-flooded; building-non-flooded; road-flooded; water; tree; ...] |
| UAVid (Lyu et al., 2020) | uavid.nl | CC BY-NC-SA 4.0 | val | 8 | [others; building; road; tree; grass; moving car; parked car; humans] |
| Kvasir-Inst. (Jha et al., 2021) | simula.no | custom | test | 2 | [others; tool] |
| CHASE DB1 (Fraz et al., 2012) | kingston.ac.uk | CC BY 4.0 | test | 2 | [others; blood vessels] |
| CryoNuSeg (Mahbod et al., 2021) | kaggle.com | CC BY-NC-SA 4.0 | test | 2 | [others; nuclei in cells] |
| PAXRay-4 (Seibold et al., 2022) | github.io | custom | test | 4x2 | [others; lungs], [others, bones], [others, mediastinum], [others, diaphragm] |
| Corrosion CS (Bianchi & Hebdon, 2021) | figshare.com | CC0 | test | 4 | [others; steel with fair corrosion; ... poor corrosion; ... severe corrosion] |
| DeepCrack (Liu et al., 2019) | github.com | custom | test | 2 | [concrete or asphalt; crack] |
| PST900 (Shivakumar et al., 2020) | github.com | GPL-3.0 | test | 5 | [background; fire extinguisher; backpack; drill; human] |
| ZeroWaste-f (Bashkirova et al., 2022) | ai.bu.edu | CC-BY-NC 4.0 | test | 5 | [background or trash; rigid plastic; cardboard; metal; soft plastic] |
| SUIM (Islam et al., 2020) | umn.edu | MIT | test | 8 | [human diver; reefs and invertebrates; fish and vertebrates; ...] |
| CUB-200 (Welinder et al., 2010) | caltech.edu | custom | test | 201 | [background; Laysan Albatross; Sooty Albatross; Crested Auklet; ...] |
| CWFID (Haug & Ostermann, 2015) | github.com | custom | test | 3 | [ground; crop seedling; weed] |

# D   MORE QUALITATIVE RESULTS

We provide more qualitative results on A-847 (Zhou et al., 2019) in Fig. 8, PC-459 (Mottaghi et al., 2014) in Fig. 9, A-150 (Zhou et al., 2019) in Fig. 10, and PC-59 (Mottaghi et al., 2014) in Fig. 11. We also further compare the results in A-847 (Zhou et al., 2019) with other methods (Ding et al., 2022a; Xu et al., 2022; Liang et al., 2022) in Fig. 12.

# E   LIMITATIONS

To evaluate open-vocabulary semantic segmentation results, we follow (Ghiasi et al., 2022; Liang et al., 2022) and compute the metrics using the other segmentation datasets. However, since the ground-truth segmentation maps involve some ambiguities, the reliability of the evaluation dataset is somewhat questionable. For example, the last row of Fig. 4 (e) exemplifies how our predictions in the mirror, "sky" and "car", as well as "plant" in between the "fence", are classified as wrong segmentation as the ground truth classes are "mirror" and "fence". Constructing a more reliable dataset including ground-truths accounting for above issue for accurate evaluation is an intriguing topic.

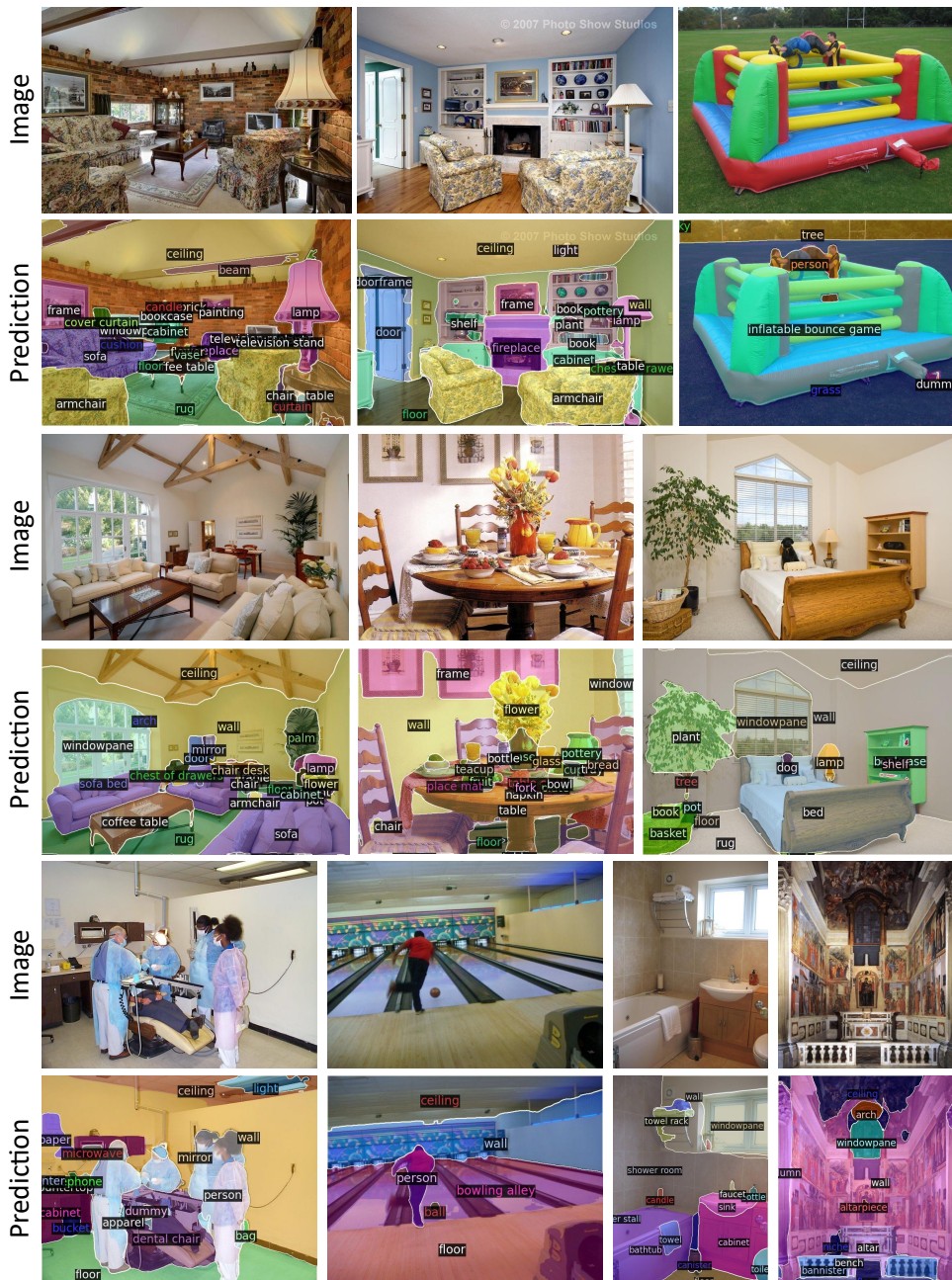

Figure 8: **Qualitative results on ADE20K (Zhou et al., 2019) with 847 categories.**

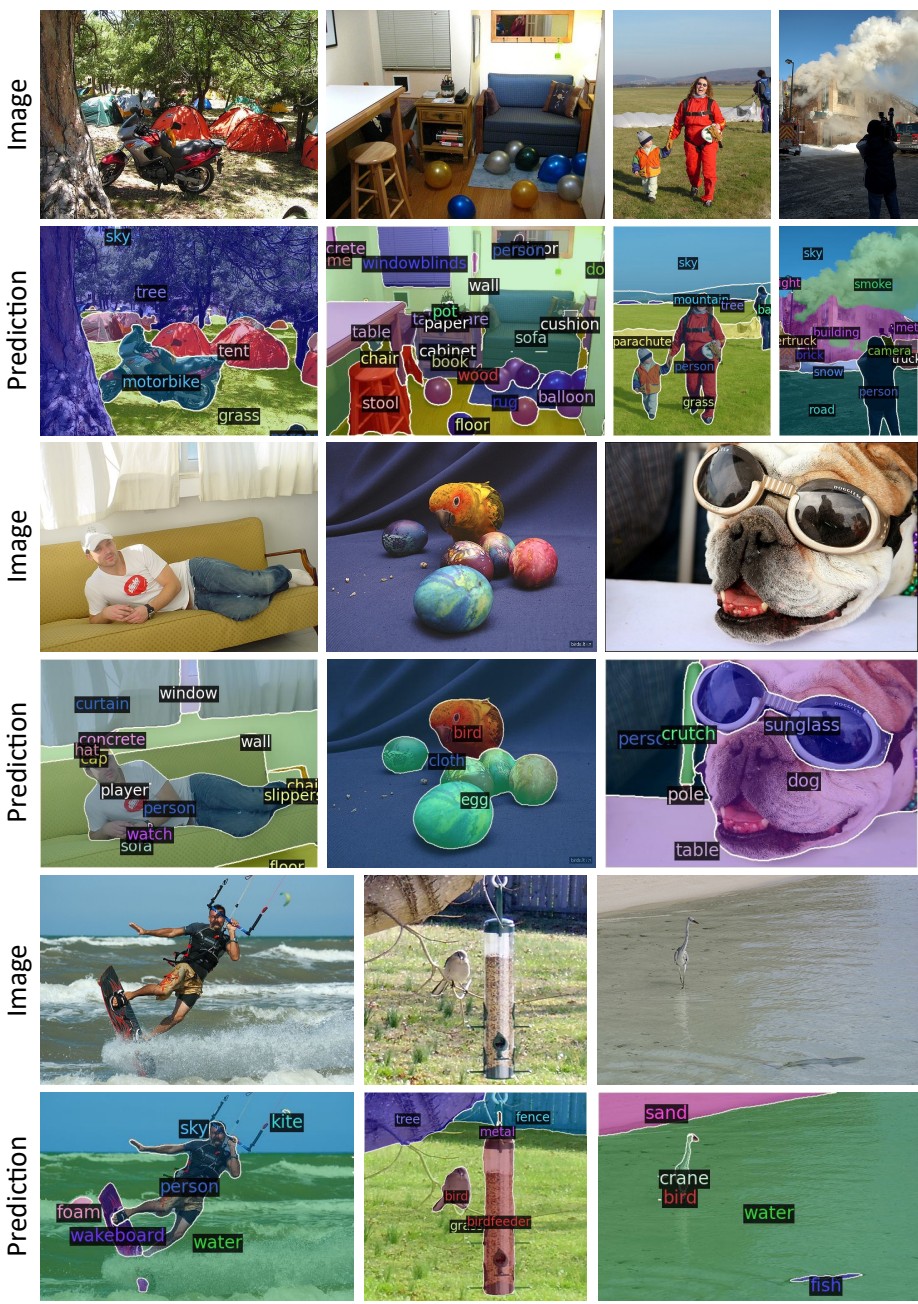

Figure 9: **Qualitative results on PASCAL Context (Mottaghi et al., 2014) with 459 categories.**

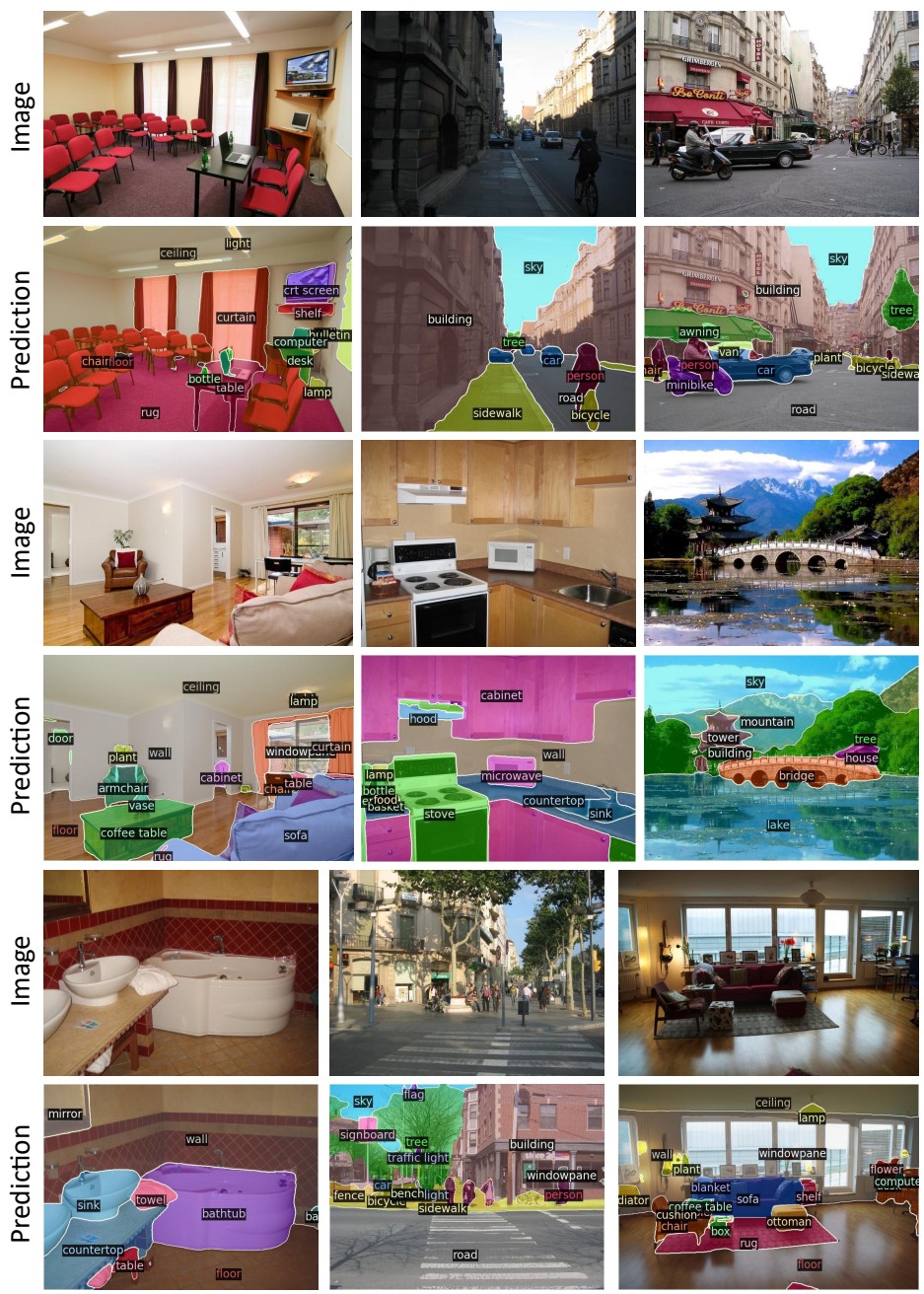

Figure 10: **Qualitative results on ADE20K (Zhou et al., 2019) with 150 categories.**

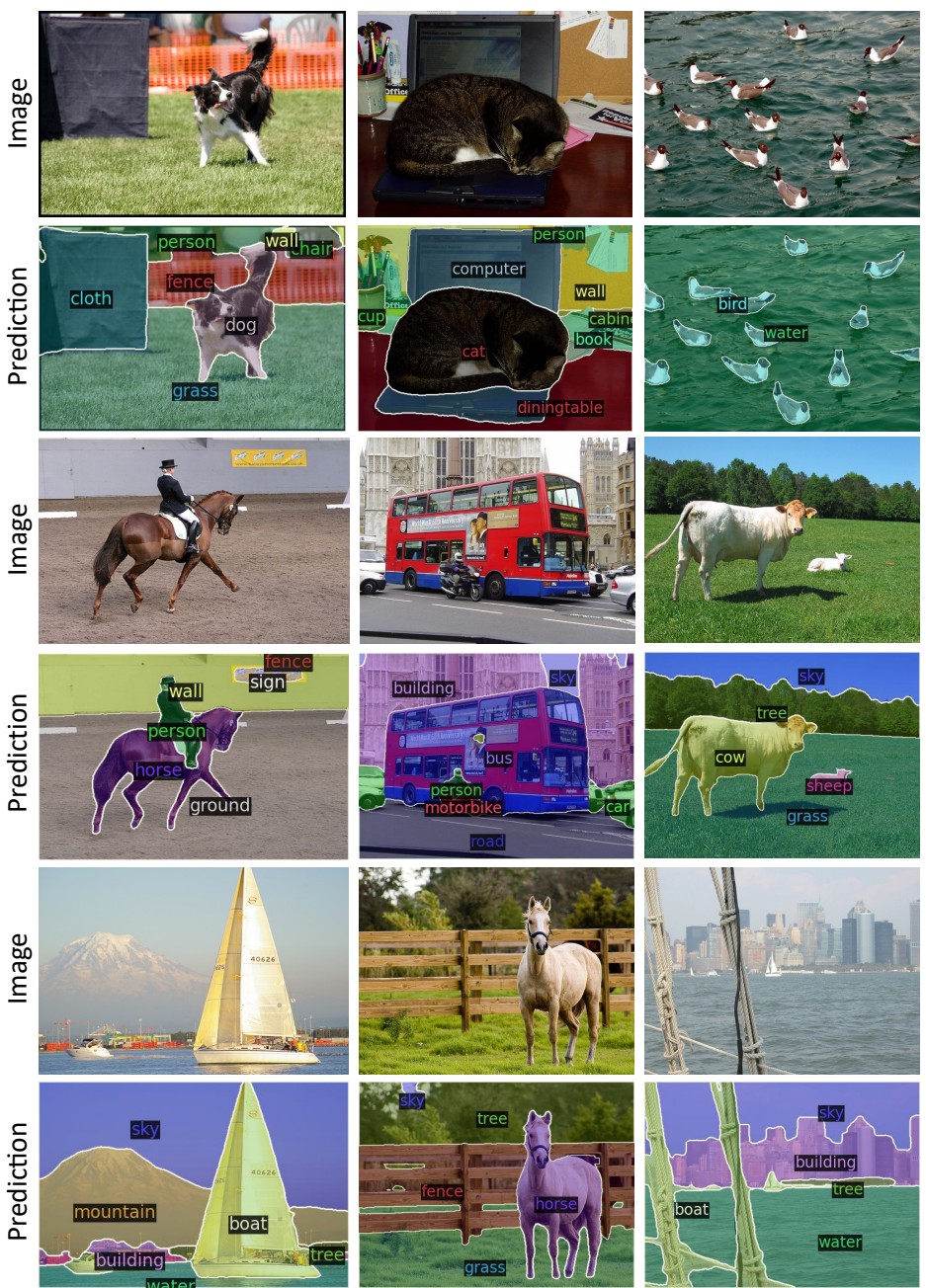

Figure 11: **Qualitative results on PASCAL Context (Mottaghi et al., 2014) with 59 categories.**

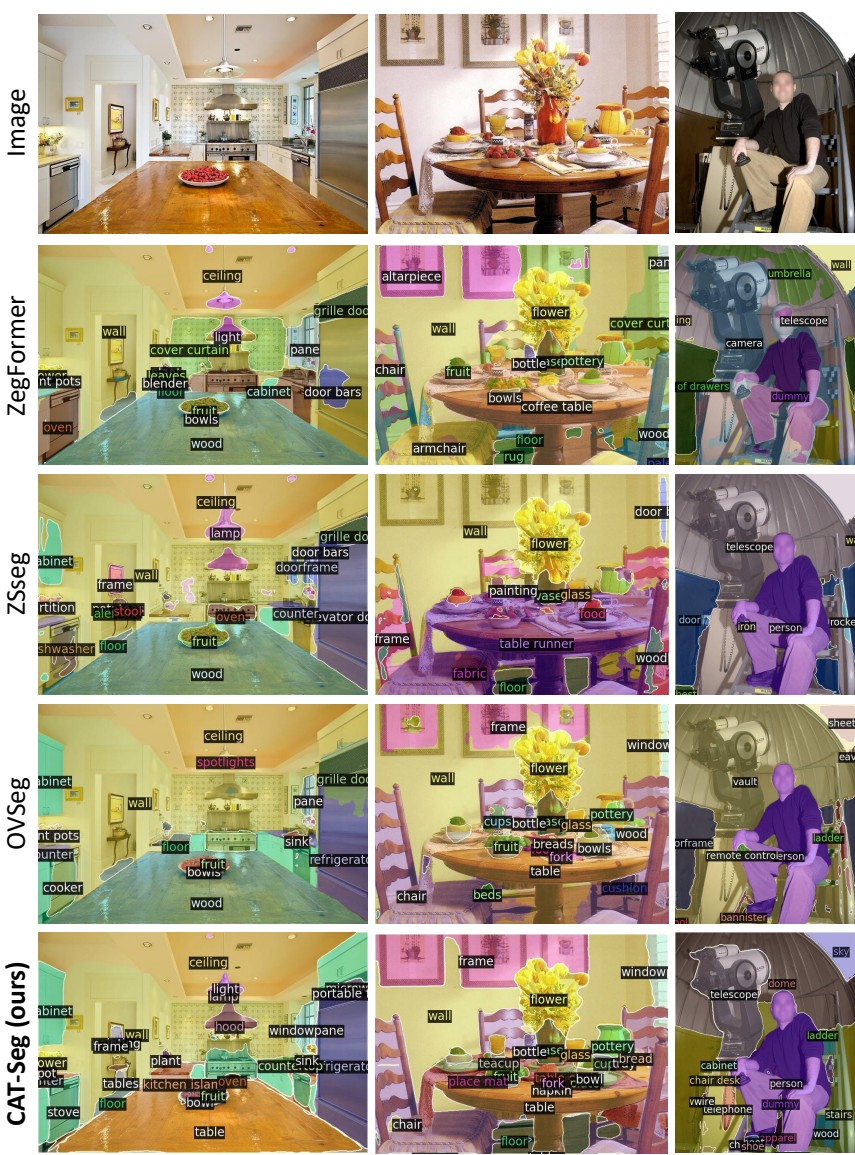

Figure 12: **Comparison of qualitative results on ADE20K (Zhou et al., 2019) with 847 categories.** We compare CAT-Seg with ZegFormer (Ding et al., 2022a), ZSseg (Xu et al., 2022), and OVSeg (Liang et al., 2022) on A-847 dataset.

