# OpenReview forum: "CAT-Seg: Cost Aggregation for Open-vocabulary Semantic Segmentation"
_ICLR.cc/2024/Conference — ICLR 2024 Conference Withdrawn Submission_

### Official Review · Reviewer_sjbY · 2023-10-31

**Soundness:** 2 fair
**Presentation:** 3 good
**Contribution:** 2 fair
**Rating:** 5
**Confidence:** 4

**Summary:**

This manuscript presents a CAT-Seg model for open-vocabulary semantic segmentation. They utilize the cosine similarity to compute the initial cost between image and text embeddings and refine it with a cost aggregation stage. As the core stage, CAT-Seg decomposes it into spatial and class aggregation, which enables the end-to-end fine-tuning for open-vocabulary semantic segmentation. Experiments on in-domain dataset, including ADE20K, PASCAL VOC, PASCAL-Context, and multi-domain MESS dataset demonstrate that CAT-Seg can well-process the unseen class.

**Strengths:**

#1 Transferring the cost aggregation from image-to-image correlation to image-to-text correspondence. The extended multi-modality cost aggregation addresses the spatial relations in images, the permutations invariance of classes and the variable number of classes that occurred in inference.

#2 Consider the challenges of variable number of categories and the unordered input arrangement, the class aggregation enables the handling of sequences of arbitrary length and the permutation to input classes.

**Weaknesses:**

#1 Though the authors state that class aggregation is permutation invariance, it is not clearly presented how the unordered input arrangement is solved? In Eq.(3), it only performs several transformer blocks and cannot prove that the function is permutation invariance. This statement also has not been verified in the experiment.

#2 Lacking the evaluation on the upsamping decoder. The authors conduct further aggregation with light-weight convolution layers, but do not further verify it in the ablation.

**Questions:**

Firstly, the author should make it clear why Eq.(3) can address the permutation invariance to the inputs. The reviewer suggest the author provide the visualization to the permuation invariance so that it can be understoodble to the reader.
Secondly, the impact of upsmapler selection can be conducted in the ablation study part as the upsampler is critical to high-resolution semantic segmentation.

---

### Official Review · Reviewer_qcKD · 2023-11-07

**Soundness:** 3 good
**Presentation:** 3 good
**Contribution:** 1 poor
**Rating:** 3
**Confidence:** 4

**Summary:**

This paper tackles open-vocabulary semantic segmentation by leveraging the cost volumes. Specifically, the correlations are obtained from text and visual information, and the spatial and class correlations are optimized, in order to refine the coarse correlation map to the finer ones, such that fine segmentation predictions can be made. Decent performance has been achieved on popular benchmarks.

**Strengths:**

1. The overall idea is straightforward and easy to understand.

2. The motivation is simple and clear: obtain finer predictions by refining the coarse correlations

**Weaknesses:**

The main weakness lies in the incremental contribution. The optimization of correlations from coarse to fine has been extensively explored in the field of semantic correspondence [1][2][3][4]. Furthermore, similar concepts have been applied to the community of segmentation [5][6].


Drawing from my experience, the success of optimizing dense correlations from coarse to fine in segmentation  [5][6] has demonstrated the effectiveness of this approach.

The difference between few-shot segmentation and open-vocabulary segmentation lies in the source of the correlation. In few-shot segmentation, the correlation is obtained from query and support images, while in open-vocabulary seg it is from the text-image pair. But the later processing pipelines are identical. Therefore, applying it to zero-shot segmentation is certainly feasible.

Moreover, the proposed spatial and class correlation modeling actually together behave like the 4-D convolutions that have been leveraged and well exploited in the aforementioned methods.


Considering the widely verified techniques employed in this paper, the method itself lacks sufficient technical contribution when compared to the aforementioned methods. Therefore, without a major improvement/change regarding the core idea of the method, I may maintain my negative rating.



References:

[1]: TransforMatcher: Match-to-Match Attention for Semantic Correspondence. CVPR 2022

[2] Learning to Compose Hypercolumns for Visual Correspondence. ECCV 2020

[3] CATs: Cost Aggregation Transformers for Visual Correspondence. Nips 2021

[4] CATs++: Boosting Cost Aggregation with Convolutions and Transformers. TPAMI 2022

[5]: Hypercorrelation Squeeze for Few-Shot Segmentation. ICCV 2021

[6]: Cost Aggregation with 4D Convolutional Swin Transformer for Few-Shot Segmentation. ECCV 2022

**Questions:**

Please respond to my questions in the weakness section.

---

### Official Review · Reviewer_3c8N · 2023-11-07

**Soundness:** 2 fair
**Presentation:** 3 good
**Contribution:** 2 fair
**Rating:** 5
**Confidence:** 4

**Summary:**

In this paper, the authors adopt a cost-aggregation mechanism to improve the open-vocabulary segmentation task. To be specific, the authors conduct the spatial- and channel-level aggregation based on the cost maps rather than the raw feature embedding from the CLIP or other open-set models, which help the model gain finer details from the cost map and optimize this cost transportation problem by the model. And they reach competitive segmentation performances on open-vocabulary settings among several benchmarks..

**Strengths:**

1. Competitive open-vocabulary segmentation performances are obtained.
2. The writing of this paper is easy to follow.

**Weaknesses:**

1. My main concern of this paper lies in the novelty. It seems that this paper applies a cost-map optimization method to model the relationship between the image embeddings and test embeddings, which have already been studied widely, such as OTA problem.
2. I think the interpretation of this paper about the cost-map is some kind of poor. Like, the authors do not demonstrate the learned embeddings of the cost-map or relationships between the image embeddings and text embeddings. It is hard to know why applying a cost-optimization following the feature embedding can improve the open-vocabulary segmentation task, assigning more accurate text labels to image pixels embeddings?
3. The reason why the authors choose different transformer blocks to implement the spatial- and channel-aggregation is not well presented and such ablative studies are missing.
4. The decoder upsampling layer proposed by this paper has already been applied as a common layer for the segmentation community and the inspiration of this part is less novel.

**Questions:**

1. In think the author should compare their method with other related cost-optimization papers on the segmentation task or the open-vocabulary task.

For example,  Learning object-language alignments for open-vocabulary object detection.

And more important thing is that this paper shares much similar idea with paper, CATs: Cost Aggregation Transformers for Visual Correspondence NeurIPS 2021. The authors should really explain more in-depth difference between these two papers. Otherwise, I think the novelty of this paper dose not meet the standard of this conference.

2. The visual demonstration of the cost map between image embeddings and text embedding should be provided to help us capture the point that cost-map here indeed does something.

3. Some ablative studies are missing as I mentioned above.

4. The introduction of decoding layer in this paper should be well demonstrate the motivation and the difference with existing works.